# Personalized Antenatal Corticosteroid Therapy and Central Nervous System Development: Reflections on the Gold Standard of Fetomaternal Therapy

**DOI:** 10.3390/diseases12120336

**Published:** 2024-12-20

**Authors:** Ivana R. Babović, Radmila Sparić, Snežana D. Plešinac, Dušica M. Kocijančić Belović, Jovana D. Plešinac, Slavica S. Akšam, Vera D. Plešinac, Giovanni Pecorella, Andrea Tinelli

**Affiliations:** 1Faculty of Medicine, University of Belgrade, 11000 Belgrade, Serbia; ivana.r.babovic@gmail.com (I.R.B.); radmila@rcub.bg.ac.rs (R.S.); snezana.plesinac@med.bg.ac.rs (S.D.P.); dusicakocijancic@yahoo.com (D.M.K.B.); slavicaaksam2012@gmail.com (S.S.A.); 2Clinic for Gynecology and Obstetrics, University Clinical Center of Serbia, 11000 Belgrade, Serbia; jplesinac@gmail.com (J.D.P.); veraplesinac@gmail.com (V.D.P.); 3Department of Obstetrics and Gynecology and CERICSAL (Centro di Ricerca Clinico SALentino), “Veris delli Ponti Hospital”, Via Giuseppina Delli Ponti, 73020 Scorrano, Lecce, Italy; giovannipecorella2690@gmail.com

**Keywords:** glucocorticoids, pregnancy, fetus, personalized antenatal corticosteroid therapy, fetal brain neurodevelopment, fetal programming

## Abstract

Background: The term “fetal programming” refers to the effects of endogenous and exogenous corticosteroids, whether received from the mother or the fetus, on brain development and the hypothalamic–pituitary–adrenal axis reset. The authors of this narrative review examine the WHO’s guidelines for prenatal corticosteroids in pregnant women who are at high risk of premature delivery. These guidelines are regarded as the best available for preventing late-life problems resulting from preterm. Methods: In order to find full-text publications published in peer-reviewed journals between 1990 and 2023 that were written in English, the authors searched PubMed, Scopus, Cochrane Library, and Web of Science. Results: The authors highlight the possible adverse long-term effects of prenatal corticosteroid medication on human brain development and function. This pharmacological feature is therapeutically significant because there is less evidence in the scientific literature regarding the potential role that the timing, mode, and dosage of exogenous steroid treatment may have in neurological illnesses down the road. Conclusions: The authors expect that these studies will shed light on the relationship between specially designed prenatal corticosteroid therapy and the molecular mechanisms underlying the prenatal programming of neurodevelopment in childhood and adulthood.

## 1. Introduction

Antenatal corticosteroid therapy (ACST) has commonly been used in the perinatal period, but this type of fetal–maternal therapy may have adverse effects on the developing fetus, cardiovascular system, and adrenal glands, and, ultimately, it may have long-term neurodevelopmental effects in preterm newborns, children, and adults [1]. Almost fifty years ago, in 1972, a trial conducted by Liggins and Howie demonstrated that the antenatal administration of glucocorticoids (GCs) to mothers “at risk of preterm birth” decreases the severity of respiratory distress syndrome (RDS) and improves the survival of preterm infants [2]. A meta-analysis of 12 randomized controlled trials on ACST was published by Crowley et al. in 1990, indicating that this therapy considerably decreased overall infant mortality, RDS, neonatal intraventricular hemorrhage (IVH), and necrotizing enterocolitis (NEC) [3]. Two years later, postnatal bronchopulmonary dysplasia (BPD) was the subject of substantial research into therapy and prevention using dexamethasone, a potent long-acting steroid with anti-inflammatory characteristics. This resulted from the trial conducted by Liggins and Howie, which examined the impact of ACST on preterm infants’ perinatal outcomes. Dexamethasone is administered in doses between 0.1 and 0.5 mg/kg/day. Its postnatal effects last between 3 and 42 days, during which preterm children are more likely to experience neurodevelopmental impairment and cerebral palsy (CP) [4,5,6]. As a result, ACST is either viewed as a fetal–maternal therapy or as the source of various contentious effects on the fetal brain’s development and the genetics of chronic illnesses in childhood and adulthood. In the paragraphs that follow, the authors summarize information highlighted by the recent scientific literature as they discuss the effects of endogenous and exogenous steroids on brain development.

### 1.1. Endogenous Steroids and Fetal Brain

The placenta is a special and separate organ connecting the mother with the fetus—two compartments that are similar, but quite distinct. Total and free maternal adrenocorticotropin hormone (ACTH) concentrations rise gradually during pregnancy. The placental synthesis of ACTH leads to a two- or three-fold increase in its concentration. Besides pneumocytes type II in pregnancy, one of the target tissues affected by ACST, in single or multiple doses, is the placental villi. The latest results confirm that ACST effects include abnormalities of villous development, placental infarcts, and decreased vascular reactivity. These findings suggest that chronic hypoxia and rapid villous maturation could be contributing factors to preterm labor [7]. Dehydro-epiandro-stenedion sulfate (DHEAS) is significantly secreted by the adrenal glands at 8–10 weeks of gestation and the fetal pituitary–adrenal axis is functional. The fetal zone (comprising most of the adrenal cortex) is the major site of steroidogenesis before birth. Precursors of fetal steroids are placental 5-pregnenolone (P5 or pregn-5-en-3ẞ-ol-20-one) and progesterone (pregn-4-en-3, 20-dion). Fetal cortisol concentrations are 5–10 times lower than the maternal concentrations [8]. Corticosteroids are essential for normal fetal brain development. In fact, two types of corticosteroid receptors are expressed in the fetal brain: mineralocorticoid receptors (MRs) and glucocorticosteroid receptors (GRs). MRs are predominantly expressed in the limbic structures (hippocampus). GRs are more diffusely distributed in the limbic system, the hypothalamic paraventricular nucleus, and the cerebral cortex. Under stressed conditions, the level of fetal cortisol rises and the occupation of GRs increases, while MRs are predominant in basal conditions [9].

### 1.2. Exogenous Steroids and Fetal Brain-Experimental Discoveries

GRs are the primary binding site for exogenous glucocorticosteroids, such as betamethasone or dexamethasone. Exogenous synthetic glucocorticosteroids likely affect brain development through altering the expression of these particular receptors in the brain; the degree of this alteration may depend on the dose, timing, and duration of ACST at the time of exposure. Dexamethasone hampers neurogenesis and causes apoptosis in the developing brain, as evidenced by animal research [10]. When the developing mouse brain is exposed to ACST, especially in high doses, the hippocampus area’s architecture and function are altered, which ultimately results in neuronal death. Antenatal dexamethasone results in significant reduction in brain weight, long-term decrease in the number of cerebellar neurons, and dose-dependent reduction in the capacity of the hippocampus [1,11]. Glucocorticoids have an impact on oligodendrocyte maturation. Dextamethasone treatment stimulates myelination during the early stages of brain development but decreases it during the late developmental stage [12]. Since “fetal programming” is a term used to describe how early-life experiences influence fetal development and the later risk of disease, prenatal stress-induced fetal programming is associated with an increased risk of preterm birth in humans and a possible risk of neurological and metabolic diseases later in life. A critical determinant of this is the regulation of fetal exposure to endogenous glucocorticoids through the placenta and exogenous glucocorticoids through ACST.

The authors of this narrative review examine the WHO’s guidelines for prenatal corticosteroids in pregnant women who are at a high risk of premature delivery. These guidelines are regarded as the best available for preventing late-life problems resulting from preterm treatment. This review’s primary objective is to present ACST as a customized treatment.

## 2. Materials and Methods

The WHO Executive Guideline Steering Group’s recommendations for the long-term application of ACST, a treatment that enhances the outcomes of preterm births, are taken into consideration in this narrative review. The most recent research data reveal the effects of ACST on neural development programming, cerebral circulation, fetal behavior (as seen by fetal breath movements), fetal neurobiogenesis, and newborn outcomes. In order to find full text publications published in peer-reviewed journals between 1990 and 2023 that were written in English, the authors searched PubMed, Scopus, Cochrane Library, and Web of Science. The terms “antenatal corticosteroids”, “fetal behavior”, “glycaemia”, “fetus”, “preterm birth”, “neonates”, “neonatal”, “intracranial hemorrhage”, “outcome”, “programming”, “neurological development”, “neurological disability”, “childhood”, “adulthood”, and “treatment” were combined to search for the relevant literature. The following studies, with or without pregnancy problems, were excluded: (a) those pertaining to premature births without ACST, and (b) manuscripts that were either not written in English or whose text was not fully available. For this narrative assessment, all manuscripts deemed by the authors to be noteworthy for the subject matter were chosen.

## 3. Results

Conventionally, courses of antenatal steroid treatment consist of four intramuscular (i.m.) administrations of 6 mg dexamethasone every 12 h, or two doses of 12 mg betamethasone given 24 h apart [13]. This course of treatment is standardized in the USA. For this reason, if a woman has not previously had an ACST course, the American College of Obstetrics and Gynecology (ACOG) does not advise giving her more than two prenatal steroid doses [14]. The odds ratio was 1.02 [95% CI, 0.81 to 1.29]; *p* = 0.84. The MACS-5 study found no statistically significant difference in the groups’ risk of death or neurodevelopmental disability: 217 of 871 children (24.9%) in the multiple-course group vs. 210 of 848 children (24.8%) in the single-course group. The study of multiple courses of antenatal corticosteroids for preterm birth at five years of age provides sufficient evidence against the use of multiple courses of ACST, indicating neither an increased nor a decreased risk of neurodevelopmental and neuropsychiatric conditions at that age [15].

These ailments include gestational diabetes mellitus, severe hypertensive disorders during pregnancy, numerous pregnancies, and fetal intrauterine development limitation.

Antenatal corticosteroid therapy (ACST) has important positive effects for short-term impacts on neonatal outcomes after preterm births, as can be seen in Table 1.

## 4. Discussion

The primary factor influencing the development of fetal organs, such as the brain, kidneys, and lungs, is the sharp rise in endogenous glucocorticosteroid levels in the fetal blood during the third trimester of gestation [16]. Fetal glucocorticosteroid levels are regulated by the placenta, and maternal glucocorticosteroid levels are regulated by the hypothalamus–pituitary axis (HPA) through negative feedback. Increased amounts of endogenous glucocorticoids in the bloodstream can have detrimental consequences on the heart, brain, metabolism, and reproductive systems when this negative feedback loop in the HPA is dysregulated [17,18].

Recently, it was demonstrated that ACST induced fetal hormonal alteration. Antenatal HPA function may show increased or decreased sensitivity to glucocorticosteroids, depending on the type of exposure. Since neural stem/progenitor cells (NSPCs) eventually influence behavior and brain development, they are particularly vulnerable to elevated levels of endogenous glucocorticosteroids. According to this research, the degree of exposure may have long-term impacts on NSPCs. A recent investigation revealed that many of the effects of exogenous glucocorticosteroids or ACST on the developing brain are positive and necessary for proper development and including NSPC proliferation, differentiation and survival, especially in preterm births [19].

In human gestation, the level of glucocorticoids in maternal circulation increases significantly starting from the 12th week of gestation, due to an increase in the release of the corticotropin-releasing hormone (CRH) from the maternal hypothalamus. During the third trimester of pregnancy, glucocorticosteroid concentration in the maternal circulation continues to increase due to CRH secretion by the placenta. At the same time, the maternal hypothalamus secretes cortisol, which increases placental CRH secretion, creating a positive feedback loop that only stops after birth. This physiologic increase in glucocorticosteroids is necessary for the maturation of fetal organs before birth in term-born infants. The enzyme 11β-hydroxysteroid dehydrogenase type-2 (HSD11B2), secreted by trophoblasts, catalyzes the conversion of active cortisol into inactive cortisone. Thus, this conversion protects the fetus from excessive endogenous maternal cortisol exposure. Fetal cortisol levels have been shown to be 10 to 13 times lower than those found in maternal circulation. Antenatally administered glucocorticosteroids take the place of maternal cortisol in promoting organ development in preterm birth. However, there are differences in their pharmacological properties. The two most common synthetic glucocorticosteroids administered to pregnant women are betamethasone and dexamethasone, which are fluorinated corticosteroids and are twenty-five times more potent than cortisol. Betamethasone and dexamethasone are also resistant to metabolism by HSD11B2. Therefore, they enter fetal circulation and pass through the placenta without hindrance, despite a closely controlled decline in HSD11B2 towards the end of pregnancy [20].

Given that ACSTs are frequently given during the third trimester of pregnancy before the body’s natural cortisol surge, and that they cross the placenta without restriction, they cause the developing fetus to activate the glucocorticoid receptor. This prevents the harmful effects of preterm birth, but may increase the risk of metabolic disease, heart problems, and neuropsychiatric disorders in newborns, children, and adults. Furthermore, elevated maternal cortisol levels after ACST administration in pregnancy are associated with more emotional disorders in girls at seven years of age. These disorders are directly related to maternal cortisol and long-term programming effects [20,21]. Betamethasone has a greater decrease in mortality and fewer side effects when compared to dexamethasone, consequently making it the current corticosteroid of choice for ACST [18]. To the best of our knowledge, there are not many studies about the relation between ACST effects and fetal HPA reactivity during the transition from childhood to adolescence and adulthood. While some studies have confirmed positive effects of ACST on decreased frequency of respiratory distress syndrome (RDS), intraventricular hemorrhage (IVH), and necrotizing enterocolitis (NEC), other studies have documented persistent disruptions in the neuroendocrine functioning of neonates with low birthweight (LBW), preterm infants and offspring [22]. Ilg et al. (2019) reported reduced basal and stress-induced cortisol concentrations in ACST-treated LBW preterm infants compared to untreated controls [23]. Finken et al. supported the finding that it is very difficult to distinguish in these infant groups between the effects of preterm and ACST on HPA dysregulation. As an independent cause for HPA dysregulation, these authors suggest ACST [24]. According to WHO recommendations in 2022, ACST may be considered as a treatment in term pregnancies [13].

Some studies, including Lammertink et al. (2021), showed attenuated cortisol levels in preterm and term ACST treated infants. This may be a reflection of the transient immaturity of the HPA in these infants [25]. Babović et al. determined that maternal corticosteroid therapy interferes with the diurnal rhythm in fetal movements, with around 50% decreased movement for 24–48 h as well as a decreased number of breathing movements in preterm births [26]. At the same time, Edelmann et al. confirmed ACST effects on major stress response systems later in life, but only in term-born infants. This constitutes the first evidence of persistent changes in HPA function, including attenuated cortisol and flatter diurnal slopes, in this group [27]. The follow-up study by Van den Bergh BRH et al. shows that a hyper-responsive HPA persists after ACST throughout late adolescence [28]. These results might be connected to ACST and a delayed onset of stress-related illnesses [23,29]. According to data from the literature, almost 80% to 90% of maternal endogenous glucocorticoids are inactivated by the placental enzyme HSD11B2.

Exogenous glucocorticosteroids pass through the placenta and directly affect the developing brain. Glucocorticoids act through their receptors (GRs). The majority of them are found in the hippocampal area. When compared to untreated controls, the hippocampal neuronal density of preterm newborns exposed to ACST was found to be lower in the human post-mortem neonatal brain [30]. Neurogenesis, gliogenesis and the production of NSPCs depend on whether ACST was given in a single dose or in multiple ones. This process indirectly contributes to cognitive and behavioral impairments observed in infants exposed to ACST in utero [31]. This hypothesis explains that the timing of neurogenesis and gliogenesis in humans is controlled and regulated by the formation of oligodendrocytes and astrocytes before and after birth [18]. Recent studies have demonstrated that maternal stress, as an epigenetic factor in late pregnancy, is related to DNA methylation of the GR gene in the fetal hippocampus. This process can be a cause of hyperactivity of the HPA. Furthermore, these findings suggest that ACST may cause persistent changes in the fetal HPA by inducing subtle neurodevelopmental disruptions as well as epigenetic modifications [32]. This instability of the fetal HPA and a more sensitive response to cortisol may persist in childhood and adolescence. Furthermore, this is a possible explanation for ACST and the fetal programming of HPA responsiveness as well as for major developmental changes and long-term vulnerability in offspring.

Assessment of fetal hemodynamic status after corticosteroid administration by Doppler ultrasonography is essential in clinical practice, especially in a sub-group of fetuses with restricted intrauterine growth. Elwany et al. reported that maternal betamethasone administration has no effect on fetal cerebral circulation. On the other hand, Henry et al. observed reduced pulsatile (PI) and resistance (RI) indices in the fetal middle cerebral artery (MCA) for 7–10 days after dexamethasone administration [33,34]. However, some researchers believe that exogenous steroids increase fetal blood pressure. This transient fetal hypertension may be a trigger for the increase in MCA PI, as a mechanism of autoregulation in order stabilize cerebral circulation [35]. Finally, Babovic et al. observed no difference between RI in cerebral circulation before and after a direct single intramuscular dexamethasone dose of 4 mg in fetuses up to 1750 g of estimated fetal weight (those below the limit were treated with 3 mg, and those above the limit with 4 mg) [36,37]. Intrauterine fetal growth restriction was excluded from all these studies [33,34,35,36,37].

The human fetal adrenal gland is active from 8 weeks of pregnancy, but adrenal cortisol is only produced in appreciable quantities after 22 weeks of pregnancy. It is common knowledge that both endogenous and exogenous steroids have a significant impact on embryonic neurodevelopment and the maturation of other organs, including the heart, the kidney, the adrenal glands, and the central nervous system [38]. The suppression of cell proliferation, DNA replication, and terminal differentiation in late gestation are the most important effects of corticosteroids on the maturation of fetal organs [39]. The expression of mdr1a P-glycoprotein (P-gp) has been confirmed during fetal brain development and blood–brain barrier (BBB) differentiation from 12 weeks of gestations. It demonstrates its most significant protective function against many agents (endogenous and exogenous steroids) during the second and the third trimester of gestation. The essential importance of this function may be efflux activity that is regulated by interactions with the caveolar endothelial cell compartment. The presence of mdr1a P-glycoprotein is also important in investigations into adult neuro-oncology [40]. Corticosteroids reach the brain and interact with two types of receptors: mineral corticosteroid receptors (MRs) and glucocorticoid receptors (GRs). Upon ligand binding in the cytoplasm, they act as transcription factors in the nucleus, either by direct interaction with DNA recognition sites (glucocorticoid response elements) or through interaction with other transcription factors, in the promoter region of target genes. These steroid receptors often demonstrate an antagonistic mode of action. Glucocorticosteroid receptors are predominantly distributed and expressed in the placenta and brain, both in glia cells and neurons, with a particular emphasis on the hippocampus. The ontogeny of corticosteroid receptor expression in the human fetal brain has not been reported. MRs have a 10-fold higher affinity for circulating endogenous glucocorticoids than GRs. Mineral corticosteroid receptors are responsible for the maintenance of basal HPA activity, while GRs mainly signal the effects during acute stress. Exogenous corticosteroids are more selective and have higher affinity for GRs. This mechanism is very important for fetal maturation and neurodevelopment [21]. An explanation for the negative impact of ACST on fetal brain development may be that it is a consequence of increased fetal corticosteroid levels, as a result of direct maternal transfer across the placenta, maternal CRH stimulation of the fetal HPA, or maternal glucocorticoid stimulated placental CRH production that activates the fetal HPA [30].

The human brain’s vulnerability to high concentrations of exogenous corticosteroids is genetically determined for each fetus. In fact, the term “fetal programming” refers to the resetting of the HPA and the impact of endogenous (maternal and fetal) and exogenous corticosteroids on brain development. A few studies have shown a connection between neurodevelopmental disorders like schizophrenia, attention-deficit/hyperactivity disorder (ADHD), antisocial behavior, increased susceptibility to post-traumatic stress disorder, anxiety disorders, learning disabilities and depression, and prenatal exposure to excess maternal corticosteroids as a result of maternal stress or exogenous maternal corticosteroids resulting from ACST [30,41,42]. These findings suggest that the manipulation of the maternal glucocorticoid milieu with ACTS is closely linked to some behavioral disorders. It is very important for clinicians that long-lasting emotional disorders and social behaviors are associated with a profound reduction in mesolimbic dopaminergic transmission in childhood and adult prenatal exposure to ACST [30,43]. It has been shown that hypoxia, increased catecholamines, and inflammatory cytokine levels can downregulate placental 11β-HSD2 in human trophoblasts, exposing the fetus to excessive amounts of endogenous glucocorticoids. Placental enzyme 11β-HSD2 insufficiency reduces hippocampal GR expression (Figure 1) [42].

Hypoxia induces the downregulation of placental 11β-HSD2 and increases fetal exposure to excessive amounts of endogenous glucocorticoids. In the adult CNS, 11β-HSD1 is key to the developmental programming of neurobehavioral dysfunctions. Low hippocampal GRs reduce glucocorticoid negative feedback and lead to exaggerated HPA responses to stress. This is a possible cause of anxiety-like behavior in adulthood [30]. In recent investigations, it has been revealed that various types of epigenetic regulation are involved in accurate gonadal differentiation in mammals. DNA methylation and histone modifications play an integral role in sex determination, which is the first step in differentiation. Epigenetic modifications regulate this process, which includes the reduced or delayed transcription of the mammalian sex-determining gene (*SRY*). Male development is highly dependent on the accurate transcription of *SRY*. *SRY* dysregulation is a potential cause of human disorders of sex development [44]. The long-term effects of prenatal glucocorticoid excess during pregnancy depend on the timing of exposure as well as on the sex of the offspring. It seems that many aspects of adverse fetal programming affect more males than females [45]. To date, long-term developmental follow-up studies in infants exposed to multiple ACST doses are limited and have produced conflicting results. In humans, this mode of therapy showed significantly lower mean birth weight, length, and head circumference than in those in the placebo group, despite no major improvements in the neonatal outcome [46]. Only six years later, in 2014, the same collaborative group (MACS-5) reported that no differences between single and multiple courses of ACST were associated with an increase or decrease in the risk of death or disability in the first 5 years of age [15].

Babović et al. observed that muscular hypotonia, as a predictor of short-term psychomotor development, occurred only in infants not exposed to maternal dexamethasone administration. However, it did not present in infants who had been exposed to this kind of treatment [46,47]. In particular, multiple antenatal doses of betamethasone resulted in impaired learning and attention disorders in 3-year-old female offspring [41]. From what has been said, it is clear that the administration of multiple doses of ACST needs to be further investigated.

## 5. Conclusions

Prenatal corticosteroid treatment alters stress-related HPA responses in both the mother and the fetus, according to the literature and evaluated studies. The responses are determined by each axis’s high sensitivity. Further studies have revealed a connection between excessive exogenous steroid exposure during pregnancy and the programming of the central nervous system, which raises an individual’s lifetime risk of behavioral, emotional, and cognitive problems. Which exogenous glucocorticoids have the best safety profile for prenatal administration is a critical research question since accurate corticosteroid levels in the fetal brain correlate with the timing, dose, and utilization in clinical practice. This research, we hope, will bolster the case for tailored, individualized ACST. Nonetheless, it is imperative to validate the relationship between tailored ACST and the molecular foundations of prenatal programming neurodevelopment in childhood and adulthood, as it will offer a new avenue for the prevention and treatment of neurodevelopmental problems in humans.

Finally, the term fetal programming implies a new area of research into the developmental causes of disease, pointing towards the in utero environment and its critical role in healthy human development, according to the fetal origins hypothesis, also known as Barker’s hypothesis. This hypothesis analyzes the relation between maternal undernutrition during the second and third trimesters of pregnancy and fetal growth restriction [48].

## Figures and Tables

**Figure 1 diseases-12-00336-f001:**
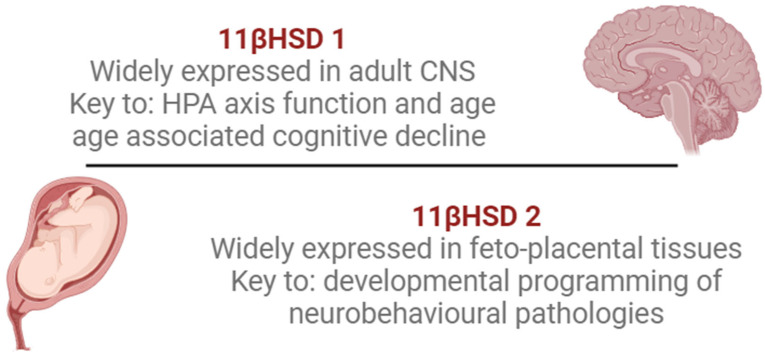
Developmental programming of neurobehavioral pathologies (Legend: HPA—hypothalamus–pituitary–adrenal; 11β-HSD1—11β-hydroxysteroid dehydrogenase type-1; 11β-HSD2—11β-hydroxysteroid dehydrogenase type-2).

**Table 1 diseases-12-00336-t001:** WHO 2022 recommendations on antenatal corticosteroid therapy for improving preterm birth outcomes.

1.0 Antenatal corticosteroid therapy is recommended for women with a high likelihood of preterm birth from 24 weeks to 34 weeks of gestation when the following conditions are met: ^+^ Gestational age assessment can be accurately undertaken ^+^ There is a high likelihood of preterm birth within 7 days of starting therapy; there is no clinical evidence of maternal infection ^+^ Adequate childbirth care is available (including capacity to recognize and safely manage preterm labor and birth) ^+^ The preterm newborn can receive adequate care (including resuscitation, kangaroo mother care, thermal care, feeding support, infection treatment and respiratory support including continuous positive airway pressure [CPAP] as needed)	Context-specific recommendation
1.1 Antenatal corticosteroid therapy should be administered to women with a high likelihood of giving birth preterm in the next 7 days, even if it is anticipated that the full course of corticosteroids may not be completed.	Context-specific recommendation
1.2 Antenatal corticosteroid therapy is recommended for women with a high likelihood of preterm birth, irrespective of whether single or multiple birth is anticipated	Context-specific recommendation
1.3 Antenatal corticosteroid therapy is recommended for women with preterm prelabour rupture of membranes and no clinical signs of infection.	Not recommended
1.4 Antenatal corticosteroid therapy is not recommended for women with chorioamnionitis who are likely to give birth preterm	Not recommended
1.5 Antenatal corticosteroid therapy is not recommended for women undergoing planned cesarean section from 34 weeks 0 days to 36 weeks 6 days.	Not recommended
1.6 Antenatal corticosteroid therapy is recommended for women with hypertensive disorders in pregnancy who have a high likelihood of preterm birth.	Context-specific recommendation
1.7 Antenatal corticosteroid therapy is recommended for women with a high likelihood of preterm birth of a growth-restricted fetus	Context-specific recommendation
1.8 Antenatal corticosteroid therapy is recommended for women with pre-gestational and gestational diabetes when there is a high likelihood of preterm birth, and this should be accompanied by interventions to optimize maternal blood glucose control.	Context-specific recommendation

ACST is not recommended when clinical signs of maternal and fetal infections are indicated: fever, maternal/fetal tachycardia, maternal leycoticosis, increased CRP, after prolonged prelabour rupture of membranes.

## Data Availability

No new data were created, since it is a narrative review of existing literature.

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
