# Peer review of "Personalized Antenatal Corticosteroid Therapy and Central Nervous System Development: Reflections on the Gold Standard of Fetomaternal Therapy"

_diseases, 2024, doi:10.3390/diseases12120336_

Round 1

Reviewer 1 Report

Comments and Suggestions for Authors

The title seemed very interesting, however, the paper has been written very poor. There are too many grammatical mistakes, I would recommend re-writing the paper. On the other hands, there is a high similarity index, that suggested to be corrected before re-submission. Finally, this review didn't find any novel aspect/finding. It is only repetition of the gaps in the literatre without any final conclusion. 

Comments on the Quality of English Language

There are too many grammatical mistakes, I would recommend re-writing the paper.

Author Response

The title seemed very interesting; however, the paper has been written very poor. There are too many grammatical mistakes, I would recommend re-writing the paper. On the other hands, there is a high similarity index, that suggested to be corrected before re-submission. Finally, this review didn't find any novel aspect/finding. It is only repetition of the gaps in the literature without any final conclusion. 

There are too many grammatical mistakes, I would recommend re-writing the paper.

Comment:  Since we agree with your remarks, we have made the necessary changes to the text. Furthermore, the fact that ACST is a widely utilized fetomaternal therapy for preventing the well-known short- and long-term sequelae of prematurity adds significance to this research. In many areas of medicine, individualized therapy is a recent development. We have some specific guidelines and regulations because every therapy, particularly during the fetal stage, may pose a risk to development and have long-term effects on the development of the newborn, adolescent, and adult stages. For the first time in the scientific literature, this review discussed ACST as individualized treatment and its connection to neurodevelopment, neurological, and psychiatric diseases and disabilities. For the first time in the scientific literature, this review discussed ACST as individualized therapy and its relationship to neurodevelopment, neurological, and psychiatric diseases and disabilities.

Finally, the authors used the publisher's editing services to carry out the suggestion for enhancing the work's technical and grammatical form.

Reviewer 2 Report

Comments and Suggestions for Authors

The manuscript reviews the current antenatal corticosteroid therapy the World Health Organization recommends. The authors conclude that research is needed into the ramifications of personalized antenatal corticosteroid therapy on neonatal and childhood neurodevelopment.

Minor suggestions:

The definition of fetal programming is not restricted to corticoids. Keller, Carrie, "Fetal Programming". Embryo Project Encyclopedia ( 2020-11-03 ). ISSN: 1940-5030 https://hdl.handle.net/10776/13180

L 72 rephrase: placental villi are one of the target tissues…

L 77 delete The

L 78 delete, as validate by recent investigations

L 81, should the delta symbol precede 5 and 4?

L 138 What does the odds ratio refer to?

Table 1.0 1 birth1, therapy n?

Table 1 1.3 Should that be is not recommended?

L 174 delete it is a well-known fact

L 179 Rephrase for clarity- the maternal hypothalamus secretes cortisol.

L 185 delete to the contrary

L 197 delete does not only, and

          change prevent to prevents

L 198 delete also

L 203 delete demonstrated

L 242 spelling methylation

L 343 change these to the

L 407 Font size on 109

Author Response

The manuscript reviews the current antenatal corticosteroid therapy the World Health Organization recommends. The authors conclude that research is needed into the ramifications of personalized antenatal corticosteroid therapy on neonatal and childhood neurodevelopment.

Comment: The authors, for the first time in the scientific literature, in this review discussed ACST as individualized therapy and its relationship to neurodevelopment, neurological, and psychiatric diseases and disabilities.

Minor suggestions:

The definition of fetal programming is not restricted to corticoids. Keller, Carrie, "Fetal Programming". Embryo Project Encyclopedia (2020-11-03). ISSN: 1940-5030 https://hdl.handle.net/10776/13180

Comment: The term fetal programming implies a new area of research into the developmental causes of disease, pointing towards the in-utero environment and its critical role in healthy human development, according to the fetal origin’s hypothesis, also known as Barker’s hypothesis. This hypothesis analyzes the relation between maternal undernutrition during the second and the third trimester of pregnancy and fetal growth restriction

L 72 rephrase: placental villi are one of the target tissues…

Comment:   English editing office (EE) changed it in: The placental synthesis of ACTH leads to a two- or three-fold increase in its concentration.

L 77 delete The

Comment:  According to the reviewer´s suggestion, we deleted it.

L 78 delete, as validate by recent investigations

Comment: English editing office changed it in: Dehydro-epiandro-stenedion sulfate (DHEAS) is significantly secreted by the adrenal glands at 8–10 weeks of gestation and the fetal pituitary-adrenal axis is functional

L 81, should the delta symbol precede 5 and 4? We change it follow recommendation

Comment: It is full term.

5-pregnenolone (P5 or pregn-5-en-3ẞ-ol-20-one) and progesterone (pregn-4-en-3, 20-dion)

L 138 What does the odds ratio refer to?

Comment: We bold significant difference between groups

The odds ratio was 1.02 [95% CI, 0.81 to 1.29]; P =.84. The MACS-5 study found no statistically significant difference in the groups' risk of death or neurodevelopmental disability: 217 of 871 children (24.9%) in the multiple-course group vs 210 of 848 children (24.8%) in the single-course group.

Table 1.0 1 birth1, therapy n?

Comment: 1 and n is a n is a typo error.

Table 1 1.3 Should that be is not recommended?

Comment: ACST is not recommends   when clinical signs of maternal and fetal infections are referred: fever, maternal/fetal tachycardia, maternal leycoticosis, increased CRP, after prolonged prelabour rupture of membranes.

L 174 delete it is a well-known fact

Comment: EE changed this sentence in: In human gestation, the level of glucocorticoids in maternal circulation increases significantly starting from the 12th week of gestation, due to an increase in the release of the corticotropin releasing hormone (CRH) from the maternal hypothalamus.

L 179 Rephrase for clarity- the maternal hypothalamus secretes cortisol.

Comment: EE rephrased it: At the same time maternal hypothalamus secretes cortisol which increases placental CRH secretion, creating a positive feedback loop that stops only after birth delete to the contrary

L 185 delete to the contrary

Comment: To the contrary, Fetal cortisol levels 185 have been shown to be 10 to 13 times lower than those found in maternal circulation.

L 197 delete does not only, and change prevent to prevents

Comment: EE changed it This prevents the harmful effects of preterm birth, but it may increase the risk of metabolic disease, heart problems, and neuropsychiatric disorders in newborns, children, and adults.

L 198 delete also

Comment: EE changed it in This prevents the harmful effects of preterm birth, but it may increase the risk of metabolic disease, heart problems, and neuropsychiatric disorders in newborns, children, and adults.

L 203 delete demonstrated

Comment: We rephrased it in to: Betamethasone has a greater decrease

L 242 spelling methylation

Comment: We spelling it into methylation.

L 343 change these

Comment: We change it in This research, we hope, will bolster the case for tailored, individualized ACST.

L 407 Font size on 109

Comment: We corrected it.

Reviewer 3 Report

Comments and Suggestions for Authors

Babovic et al. shed light on the relationship between specially designed prenatal corticosteroid therapy and the molecular mechanisms underlying the prenatal programming of neurodevelopment in both childhood and adulthood. This intriguing review highlights evidence from the literature and evaluated studies, demonstrating that antenatal corticosteroid therapy (ACST) affects stress-related HPA axis responses in both the mother and fetus. The authors also propose the potential for personalized ACST as a strategy to reduce an individual’s lifetime risk of behavioral, emotional, and cognitive disorders. However, there are several issues that need to be addressed to further enhance the manuscript.

1.     It is well-known that dexamethasone (DEX), even at high doses, poorly penetrates the brain through the blood-brain barrier (BBB) due to the presence of mdr1a p-glycoprotein expressed on the apical membranes of endothelial cells in the BBB. In the case of ACST using DEX, does DEX penetrate the brain via the fetal BBB, potentially due to its immaturity? If so, the authors should discuss the complex characteristics of DEX in greater detail.

2.     Maternal experiences, such as stress including ACST, are associated with a range of neurodevelopmental and metabolic diseases, some of which have been observed to persist into the second and third generations. The mechanisms through which factors such as ACST contribute to disease development likely involve a complex interplay between the maternal environment, placental changes, and the epigenetic programming of the embryo. While there has been growing recognition and exploration of the epigenome in determining disease risk, little is known about the role of embryo sex in epigenetic regulation. The authors are strongly encouraged to address this point in their discussion.

Author Response

Babovic et al. shed light on the relationship between specially designed prenatal corticosteroid therapy and the molecular mechanisms underlying the prenatal programming of neurodevelopment in both childhood and adulthood. This intriguing review highlights evidence from the literature and evaluated studies, demonstrating that antenatal corticosteroid therapy (ACST) affects stress-related HPA axis responses in both the mother and fetus. The authors also propose the potential for personalized ACST as a strategy to reduce an individual’s lifetime risk of behavioral, emotional, and cognitive disorders. However, there are several issues that need to be addressed to further enhance the manuscript. 1. It is well-known that dexamethasone (DEX), even at high doses, poorly penetrates the brain through the blood-brain barrier (BBB) due to the presence of mdr1a p-glycoprotein expressed on the apical membranes of endothelial cells in the BBB. In the case of ACST using DEX, does DEX penetrate the brain via the fetal BBB, potentially due to its immaturity? If so, the authors should discuss the complex characteristics of DEX in greater detail. Comment: We found this data in literature that the expression of mdr1a P-glycoprotein (P-gp) has been confirmed during fetal brain development and blood–brain barrier (BBB) differentiation from 12 weeks of gestations. It demonstrates its most significant protective function against many agents (endogenous and exogenous steroids) during the second and the third trimester of gestation. The essential importance of this function may be efflux activity that is regulated by interactions with the caveolar endothelial cell compartment. Reference: Virgintino, D; Errede, M; Girolamo, F; Capobianco, C; Robertson, D; Vimercati, A; Serio, G; Benedett DA; Yonekawa, Y; Frei, K; Roncali, L. Fetal Blood-Brain Barrier P-Glycoprotein Contributes to Brain Protection During Human Development, Journal of Neuropathology and Experimental Neurology 2018;(67): 50–61. 2. Maternal experiences, such as stress including ACST, are associated with a range of neurodevelopmental and metabolic diseases, some of which have been observed to persist into the second and third generations. The mechanisms through which factors such as ACST contribute to disease development likely involve a complex interplay between the maternal environment, placental changes, and the epigenetic programming of the embryo. While there has been growing recognition and exploration of the epigenome in determining disease risk, little is known about the role of embryo sex in epigenetic regulation. The authors are strongly encouraged to address this point in their discussion. Comment: In recent investigations, it has been revealed that various types of epigenetic regulation are involved in accurate gonadal differentiation in mammals. DNA methylation and histone modifications play an integral role in sex determination, which is the first step of differentiation. The epigenetic modifications regulate this process which includes reduced or delayed of the transcription of the mammalian sex-determining gene, (Sry). Male development is highly dependent on the accurate transcription of Sry. SRY dysregulation is a potential cause of human disorders of sex development Reference: Miyawaki, S; Tachibana, M. Chapter Seven - Role of epigenetic regulation in mammalian sex determination. In (Ed): Blanche C. Current Topics in Developmental Biology, Academic Press, 2019; (134):195-221. The long-term effects of prenatal glucocorticoid excess during pregnancy depend on the timing of exposure as well as on the sex of the offspring. It seems that many aspects of adverse fetal programming affect more males than females. Reference: Dunn, G.A.; Morgan, C.P.; Bale, T. L. Sex-specificity in transgenerational epigenetic programming. Hormones and Behavior, 2011, 59, 290–295.